# Diagnostic Use of Selected Metalloproteinases in Endometrioid Ovarian Cancer

**DOI:** 10.3390/biomedicines13092143

**Published:** 2025-09-02

**Authors:** Ewa Gacuta, Aleksandra Kicman, Paweł Ławicki, Michał Ławicki, Monika Kulesza, Paweł Malinowski, Marcin Chlabicz, Monika Zajkowska, Sławomir Ławicki

**Affiliations:** 1Department of Perinatology, University Clinical Hospital of Bialystok, 15-276 Białystok, Poland; 2Department of Psychiatry, Medical University of Białystok, 15-272 Białystok, Poland; olakicman@gmail.com; 3Department of Population Medicine and Lifestyle Diseases Prevention, University of Białystok, 15-269 Białystok, Poland; pawellawicki04@gmail.com (P.Ł.); mlawicki@icloud.com (M.Ł.); monika.kulesza@sd.umb.edu.pl (M.K.); slawomir.lawicki@umb.edu.pl (S.Ł.); 4 Department of Oncological Surgery, Bialystok Oncology Center, 15-027 Białystok, Poland; pawelmalinowski1981@gmail.com; 5Department of Urology, Ministry of Internal Affairs and Administration Hospital, 15-471 Białystok, Poland; chlabicz@outlook.com; 6Department of Neurodegeneration Diagnostics, Medical University of Białystok, 15-269 Białystok, Poland; monika.zajkowska@umb.edu.pl

**Keywords:** EnOC, cancer prevention, early diagnosis, MMP-3, MMP-7, MMP-10, MMP-11, MMP-26

## Abstract

**Background/Objectives:** Endometrioid ovarian cancer (EnOC) is a late-diagnosed gynecological cancer with limited diagnostic methods that, when detected at an early stage, has a good prognosis. This study is the first to evaluate the plasma concentrations and diagnostic utility of selected metalloproteinases as new biomarkers for EnOC. **Methods**: The study group consisted of 50 newly diagnosed, untreated patients with EnOC; the control group consisted of 25 patients with endometrial cysts, 25 patients with serous cysts, and 50 healthy women. Selected matrilysins and stromelysins were determined by means of immunoenzymatic assay (ELISA) and routine markers (CA125 and HE4) using the chemiluminescence (CMIA) method. **Results:** Higher levels of MMP-7, MMP-10, MMP-11, and MMP-26 were found in patients with EnOC when compared to healthy women. Concentrations of MMP-7, MMP-10, and MMP-11 were higher in women with EnOC when compared to benign lesions (BL). The highest SE (98.55%), NPV (95.41%), ACC (57.58%), and AUC (0.9658) values were obtained for MMP-7. High values of diagnostic parameters were also obtained for MMP-11 and MMP-26. **Conclusions:** These results suggest the usefulness of MMP-7, MMP-26, and MMP-11 in the diagnosis of EnOC as new biomarkers in this pilot study.

## 1. Introduction

Ovarian cancer (OC) is one of the most common gynecological cancers. According to global estimates, in 2022 alone, 324,398 new cases and 206,839 deaths were reported worldwide [1,2,3]. The disease is one of the more insidious types of cancer because, in its early stages, it rarely gives clear symptoms. Most often, they are nonspecific and can be attributed to various, more common health problems. Patients are most often diagnosed in advanced stages of the disease, when metastases in distant organs are already present [1,2,3,4]. Unfortunately, diagnosis in the early stages of the disease is very difficult because there is no effective screening test available. The gold standard nowadays is histopathological examination and non-invasive methods, primarily clinical examinations, transvaginal ultrasounds and laboratory tests, including a determination of markers, mainly CA 125, HE 4 alone or in the ROMA algorithm [5,6,7]. However, these methods have numerous limitations, especially when it comes to CA 125 and HE 4, as their concentrations are influenced by a number of physiological and pathological processes within a woman’s body [1,2].

High-grade serous carcinoma is the most common type of ovarian cancer, accounting for approximately 75% of epithelial ovarian cancers. In contrast, endometrioid ovarian carcinoma (EnOC) constitutes about 10% of epithelial ovarian cancer diagnoses. This is currently a poorly studied type of cancer, and it has been postulated that the precursor lesion for this type of malignant change is endometriosis, suggesting a different etiological pathway compared to other ovarian cancer subtypes [8]. Women with Lynch syndrome are also at an increased risk of developing ovarian endometrioid cancer [9]. Despite generally being associated with a more favorable prognosis than high-grade serous carcinoma, EnOC can exhibit resistance to standard chemotherapy in advanced stages. There is a need for improved therapeutic approaches personalized to this subtype [8,10]. EnOC is a distinct histological subtype with unique molecular and clinical characteristics, distinguishing it from other epithelial ovarian cancers such as serous or clear cell carcinoma [1]. Studying EnOC separately allows for a better understanding of its specific pathogenesis and behavior. Therefore, it is expedient to introduce new methods that enable more accurate and rapid diagnosis of EnOC. Such methods include the determination of new biomarkers from peripheral blood. Currently, the number of compounds with high potential in the diagnosis of ovarian cancer is steadily increasing, including CA-125, HE4, mesothelin, and several matrix metalloproteinases [11,12,13,14]. These markers have been investigated for their diagnostic utility, alone or in combination, to improve early detection and patient stratification. Matrix metalloproteinases (MMPs) are enzymes that degrade extracellular matrix components, playing a crucial role in tissue remodeling, inflammation, and tumor progression. They are of particular interest because their dysregulation has been linked to cancer development, metastasis, and other pathological processes, making them potential diagnostic biomarkers and therapeutic targets in oncology and metabolic diseases [1,2].

MMPs belong to a group of proteolytic enzymes whose activity depends on zinc ions. Their increased activity is mostly associated with the initiation and progression of many diseases, but a special role is attributed to them in the process of carcinogenesis. At the initiation stage, they are associated with the induction of genomic instability, whereas at subsequent stages, their activity is associated with an increase in the frequency of the proliferation, invasion, and migration of tumor cells and the induction of angiogenesis [2]. In advanced stages of cancer, they are associated with metastasis [1,2,15]. In the course of OC, the first studies have already demonstrated the usefulness of these enzymes in the diagnosis of this type of cancer [1,2,16,17]. Therefore, the aim of the present study was to evaluate the concentrations and diagnostic utility of selected matrilysin and stromelysin enzymes in EnOC as part of a pilot study concerning a broader group—EOCs. We would like to note that this is the first study that includes selected metalloproteinases as new diagnostic markers in a rare histological type of ovarian cancer, i.e., endometrioid ovarian cancer. We hope that our results will represent an important step towards improving the early detection of this rare malignancy by investigating the potential role of these enzymes as reliable biomarkers that may be useful in detecting malignant ovarian lesions.

## 2. Materials and Methods

Table 1 summarizes the groups of patients who participated in the experiments.

The study included 50 patients with endometrioid ovarian cancer, classified according to the FIGO staging system into one of two groups—stage I–II (early stages) and stage III–IV (late, advanced stages). At the in-hospital diagnostic stage, histopathological evaluation has been carried out, which was the basis for the study group inclusion. Clinical, blood, X-ray, and ultrasound examinations were performed before any treatment was initiated. All patients underwent computed tomography (CT) and magnetic resonance imaging (MRI) of the pelvic organs in accordance with international clinical guidelines. Patients who were eligible for the study had not previously received radiotherapy nor chemotherapy. Due to the high levels of HE4 observed in patients with renal failure, women with this condition were excluded from the study.

The control group consisted of 25 patients with endometrial cysts, 25 with serous cysts, and 50 healthy women. Patients with ovarian endometrioid cysts were treated at the Department of Gynecology of the University Clinical Hospital in Bialystok and at the University Oncology Center of the University Clinical Hospital in Bialystok from 2021 to 2025. Histopathological examination was performed in patients during the in-hospital treatment stage. Healthy female volunteers were selected based on their consultations with both a family physician and gynecologist at the University Clinical Hospital in Bialystok. Volunteers who qualified for the study underwent annual preventive examinations: laboratory tests, cervical cytology, and abdominal ultrasounds. Patients with a positive history of benign and malignant gynecological disease were not eligible for this study. Before submitting to blood tests, female volunteers underwent gynecological as well as ultrasound examinations.

The study was conducted in accordance with the Declaration of Helsinki. The protocol was approved by the local Ethics Committee of the Medical University of Bialystok (committee approval number: APK.002.420.2021 (approval date 21 October 2021 and 12 December 2024)). All patients gave their informed consent to participate in the study. A medical history was also obtained from all study participants. The course of the experiment is represented by the flowchart in Figure 1.

### 2.1. Biochemical Assays

The blood samples were collected into tubes containing lithium heparin as an anticoagulant. Within one hour after collection, the blood was centrifuged at a speed of 1810× *g* for 10 min to obtain plasma, which was then aliquoted and stored at −81 °C until assayed. Determination of selected MMPs was carried out using the immunoenzymatic method (ELISA). The reagents used were from the following: R&D Systems—MMP-3 (Human Total MMP-3 DuoSet ELISA, Cat. no. DY513, Minneapolis, MN, USA), MMP-7 (Human Total MMP-7 DuoSet ELISA, Cat. no. DY907, Minneapolis, MN, USA), MMP-10 (Human Total MMP-10 DuoSet ELISA, Cat. No. DY910, Minneapolis, MN, USA), ElabScience—MMP-11 (Human MMP-11 (Matrix Metalloproteinase 11) ELISA Kit, Cat. E-EL-H1443, Wuhan, China), and Abbkine—MMP-26: Human Matrix metalloproteinase-26 (MMP26) ELISA Kit, Cat. no. KTE61590, Atlanta, GA, USA). The precision of the kits was set by the manufacturer—MMP-3: 6.1% (intra-assay), 7% (inter-assay); MMP-7: 3.4% (intra-assay), 4.1% (inter-assay); MMP-10: 3.7 (intra-assay), 4.1 (inter-assay); MMP-11: 5.6% (intra-assay), 5% (inter-assay); and MMP-26: <9% (intra-assay), <11% (inter-assay). Standards and samples were applied to the plate in duplicate to ensure analytical reliability. The precision of the assays was established according to the manufacturers specifications. Absorbance readings were performed using a wavelength of 450 nm with a correction wavelength set at 540 nm. Routine markers HE4 and CA125 were measured using a chemiluminescent microparticle immunoassay (CMIA) (Abbott, Chicago, IL, USA)—with the use of ROCHE Diagnostics reagents (Roche Elecsys HE4 and Roche Elecsys CA125 II, Roche Diagnostics GmbH, Mannheim, Germany).

### 2.2. Statistical Analysis

Statistical analysis was performed using PQStat Software (v.1.8.4.162, Poznan, Po-land). Graphical processing was performed using GraphPad Prism Software (v. 9.1.1 (225), San Diego, CA, USA). The normality of the distribution was assessed using the Shapiro–Wilk test with Lilliefors correction—the tests showed significant deviations from the normal distribution. That is why a statistical analysis was performed using non-parametric tests. To evaluate statistical differences between the independent groups, the Kruskall–Wallis test and the Mann–Whitney U test were used, while when comparing multiple groups, the Mann–Whitney U test with Holm–Bonferroni correction for multiple queries or the Dwass–Steele–Critchlow–Fligner post hoc test were used.

Medcalc statistical calculators were used to evaluate the diagnostic properties of parameters such as sensitivity (SE), specificity (SP), positive predictive value (PPV), negative predictive value (NPV), and accuracy (ACC). The analysis was based on the area under the ROC curve (AUC) and optimal cut-off points were determined with maximization of the Youden index value for the cancer–control differential test, which were as follows: 2.197 ng/mL (MMP-7), 7550 pg/mL (MMP-26), 8803.23 pg/mL (MMP-3), 59.58 pg/mL (MMP-10), 920.56 pg/mL (MMP-11), 30.185 U/mL for CA 125 and HE4 (67.11 U/mL)

## 3. Results

### 3.1. Concentrations of Selected Matrilysins and Stromelysins in Patients with Ovarian Endometrioid Carcinoma, Patients with Ovarian Cysts, and Healthy Women

The concentrations of the tested parameters—MMP-7, MMP-26, MMP-3, MMP-10 and MMP-11—obtained in blood plasma from patients with *ovarian endometrioid carcinoma*, patients with *ovarian cysts,* and healthy women are presented in Table 2 and Table 3 and Figure 2, Figure 3, Figure 4, Figure 5 and Figure 6.

MMP-7 concentrations were highest in the EnOC group (median: 5.64 ng/mL) compared to the BL group (2.78 ng/mL; *p* < 0.0001), HW group (0.95 ng/mL, *p* < 0.0001), EC group (2.80 ng/mL, *p* = 0.001), and SC group (2.94 ng/mL, *p* < 0.0001). At the same time, there was a statistically significant difference between MMP-7 concentrations in the BL and HW groups (*p* < 0.0001). Statistically significant differences were also shown between the HW and both benign groups: EC and SC (both *p* < 0.0001) (Figure 2).

MMP-26 concentrations were statistically lower in the EnOC group (medtian: 9425 pg/mL) than in the BL group (median: 10,985 pg/mL) (*p* < 0.0001) but higher than the HW group (median: 7163 pg/mL; *p* < 0.0001). Similarly, higher MMP-26 concentrations were found in the BL patients group compared to the HW group (*p* < 0.0001). There were also statistically significant differences between the HW and both benign groups: EC group (median: 10,825 pg/mL; *p* < 0.0001) and SC group (median: 12,169 pg/mL, *p* < 0.0001) (Figure 3).

Unfortunately, there were no statistically significant differences between the tested groups in MMP-3 concentrations (EnOC median: 12,106.47 pg/mL; BL: 10,298 pg/mL; HWL: 8359 pg/mL; EC: 10,235 pg/mL; SC: 10,683 pg/mL) (Figure 4).

The highest MMP-10 concentrations (similarly to MMP-26) were found in the BL group (median: 208.80 pg/mL). This result was significantly higher compared to the EnOC group (160.35 pg/mL; *p* < 0.0001) and the HW group (59.51 pg/mL; *p* < 0.0001). Statistically significant differences were also found between the EnOC group and HW group, as well as between the EnOC group and both benign subgroups: EC group (160.35 pg/mL) and SC group (113.80 pg/mL) (both *p* < 0.0001). Similarly to the EnOC group, HW group also demonstrated significant results when compared to the divided benign lesion subgroups (Figure 5).

For MMP-11, the highest concentration was found in the EnOC group (median: 2733.13 pg/mL) compared to the BL group (2940 pg/mL; *p* < 0.001) and HW group (4400 pg/mL; *p* < 0.0001). At the same time, MMP-11 concentrations in the BL group were lower than in the HW group (*p* < 0.005). Higher concentrations were also found in the EnOC patients compared to the benign subgroups, with the EC (233.13 pg/mL; *p* < 0.0001) and the SC (200.0 pg/mL, *p*< 0.0001) (Figure 6).

Concentrations of both HE4 and CA125 were highest in the EnOC group (CA125 median: 396.61 U/mL; HE4 median: 338.01 U/mL) compared to the BL group (23.38 U/mL; 52.55 U/mL; *p* < 0.0001), the HW group (16.9 U/mL; 39.51 U/mL; *p* < 0.0001), the EC group (24.98 U/mL; *p* < 0.0001; 53.20 U/mL; *p* < 0.0001), and the SC group (19.00 U/mL; *p* < 0.0001; 51.47 U/mL) (Figure 7).

### 3.2. Evaluation of Correlation Using Spearman’s Method

Spearman’s method was used to analyze potential correlations. No significant correlations were shown in the group of patients with endometrioid ovarian cancer and the healthy women group. However, in the BL total, EC, and SC groups, statistically significant positive and negative correlations were found and are presented in Table 4. Particularly strong correlations are shown in the SC group, between MMP-26, MMP-3, MMP-10, MMP-11, and routine markers.

### 3.3. Diagnostic Criteria of Tested MMPs, CA 125, and HE4

Table 5 contains the diagnostic criteria—diagnostic sensitivity (SE), diagnostic specificity (SP), positive and negative predictive value (PPV and NPV), and accuracy (ACC)—in patients with ovarian endometrioid carcinoma.

The highest value of diagnostic sensitivity (SE) was obtained for MMP-7 (98.55%), which exceeded the SE values obtained for the routine markers CA125 (96.00%) and HE4 (80%). Equally high SE values were also obtained for MMP-26 (94%) and MMP-10 (92%).

In the case of diagnostic specificity (SP), again, the highest value from among the MMPs tested was obtained for MMP-7 (86%); similar values were also obtained for MMP-11 (82%). The SP values obtained for routine markers were significantly lower (CA125: 47.95%; HE4: 52.05%) than for all MMPs tested.

The highest value of positive predictive value (PPV) from all tested MMPs was obtained for MMP-7 (87.72%); comparable values were found for MMP-26 (70.15%) and MMP-10 (70.77%). However, those values were lower in comparison with those obtained for CA125 and HE4.

The next parameter determined was the negative predictive value (NPV) in which MMP-7 acquired the highest value from tested MMPs (95.41%). Also, a high value was obtained for MMP-26 (90.91%) and CA125 (96.08%).

The last of the parameters tested was accuracy (ACC). In this case, the best value for this parameter was obtained by MMP-10 (77.00%) and MMP-26 (77.00%).

### 3.4. Evaluation of the Diagnostic Power of Tests by ROC Function

The diagnostic power of the tests was determined by evaluating the area under the ROC curve (AUC), which made it possible to determine whether a given test can completely distinguish between a healthy individual and a person with cancer. A diagnostically ideal test allows the complete differentiation of a healthy person from a sick person, with a sensitivity value of 100% and specificity of 100%. In such a case, the Receiver Operating Characteristic (ROC) curve would align entirely along the *Y*-axis, indicating flawless discrimination, and the area under the curve (AUC) would be equal to 1. Conversely, a diagnostically ineffective test fails to distinguish between healthy and diseased populations, resulting in an ROC curve that approximates the diagonal line at a 45-degree angle relative to the *X*-axis with the AUC value of 0.5. The results of the ROC-AUC curve analysis are shown in Table 6.

The highest AUC value among the tested MMPs was obtained for MMP-7 (0.9658) and MMP-11 (0.7926). The AUC value obtained for MMP-7 exceeded that of HE4 (0.9247) and was comparable to the AUC for CA125 (0.9758). All tested parameters had AUCs of more than 0.5, indicating their diagnostic usefulness. The graphical presentation of the ROC curve for the studied parameters is presented in Figure 8.

## 4. Discussion

Endometrioid ovarian cancer (EnOC) represents a distinct subtype of epithelial ovarian cancer, both in terms of its molecular profile and clinical behavior. It is notably associated with endometriosis and demonstrates unique pathophysiological features compared to high-grade serous ovarian carcinoma. Despite growing interest in ovarian cancer research, most studies have predominantly focused on the more prevalent subtypes, particularly high-grade serous carcinoma. In contrast, EnOC, which accounts for only approximately 10% of epithelial ovarian cancer cases, remains relatively understudied, and its optimal management is still not well defined. In this context, we aimed to investigate the role of selected matrix metalloproteinases (MMPs) in the pathogenesis of ENOC, with a particular focus on their diagnostic relevance. We also sought to evaluate how the behavior and potential utility of these MMPs in detecting EnOC compares to their established roles in other, more common ovarian cancer subtypes through comparison of our results with those of other authors in the available literature. This approach may provide insight into whether MMPs have higher or lower diagnostic value specifically in the context of EnOC and help identify molecular features that could inform future targeted diagnostic or therapeutic strategies. Currently, the diagnosis of this type of cancer is based on non-specific methods, including transvaginal ultrasound and serum biomarker determinations, such as HE4 and CA125, either alone or combined in the ROMA algorithm. In the case of classical markers, their concentration is affected by a number of factors related to the physiology and pathology of the female reproductive system [1,2]. Therefore, the search for a novel, highly effective screening test for the early detection of endometrioid ovarian cancer should be of utmost importance. Circulating blood goes to and from the tumor microenvironment by which it changes its protein and enzyme profile, thus making plasma a valuable material for the search for new biomarkers [1,2]. Therefore, the aim of the present study was to evaluate the concentrations and diagnostic utility of selected matrilysin and stromelysin enzymes in the diagnosis of endometrioid ovarian cancer in comparison with women with endometrial cysts of the ovary and healthy women. All biomarker assessments were performed prior to any therapeutic intervention to eliminate potential effects of treatment on MMP levels.

The presence of MMP-7, MMP-26, MMP-3, MMP-10, and MMP-11 expression has been previously confirmed in biopsy material from ovarian cancer patients [18]. The overexpression of MMPs is mostly associated with increased mutational potential (MMPs enter the cell nucleus and degrade enzymes responsible for DNA repair), increased proliferation, migration, invasion of tumor cells, stimulation of angiogenesis, and metastasis formation [18,19,20]. Expression was significantly elevated in patients with endometrioid ovarian carcinoma compared to women with benign lesions and healthy controls. These findings suggest that endometrioid ovarian carcinoma cells produce substantial quantities of MMP-7 and MMP-11, leading to their increased concentration in the bloodstream.

In our research, we showed that patients with EnOC have higher levels of MMP-7 (5.64 ng/mL) compared to healthy patients (0.95 ng/mL) and those with benign lesions (2.78 ng/mL). Our results agree with the work of Będkowska et al. [17], whereby the authors found higher levels of MMP-7 in a group of patients with mixed ovarian cancer (EnOS accounted for 46% of all cases; median: 5.60 ng/mL), compared to benign lesions and healthy women (BL: 3.18 ng/mL; HW: 3.25 ng/mL). Also, the work of Acar et al. [21] showed higher plasma MMP-7 levels in women with OC (10.24 ng/mL) compared to healthy women (3.29 ng/mL); however, this study did not involve EnOC but rather other EOCs. MMP-7 is an enzyme responsible for the degradation of extracellular matrix components, including collagen, laminin, and elastin, which facilitates tumor invasion and metastasis. Elevated MMP-7 levels in the blood of patients with EnOC may indicate increased activity of proteolytic processes associated with tumor progression. High MMP-7 levels may also reflect the tumor’s increased potential to invade surrounding tissues and spread. Furthermore, in a diagnostic context, elevated MMP-7 levels may indicate the presence of malignant lesions, differentiating them from benign lesions, making MMP-7 a promising marker for supporting the diagnosis of EnOC.

Interestingly, both Zhai et al. [22] and Sillanpää et al. [23] revealed that, in endometrioid tumors, a high percent of tissue with MMP-7 overexpression and an intense MMP-7 signal were substantially related with nuclear positivity of b-catenin. Thus, the presence of MMP-7 in cells correlates with the presence of β-catenin in the nucleus. β-catenin regulates the cell cycle, cell adhesion, and signal transduction within the Wnt pathway, potentially impacting cancer behavior. Its high activity, particularly in the cell nucleus, may be linked to earlier cancer development as it stimulates the production of genes involved in proliferation, differentiation, and cell development. However, on the basis of the Gene Expression Omnibus, Cancer Biomedical Informatics Grid, and The Cancer Genome Atlas cancer datasets, the MMP7 gene was not associated with overall survival rate in patients with ovarian cancer [24]. Furthermore, within the course of different cancers such as breast cancer, elevated levels of MMP-7 have been found in cancer patients compared to healthy women [25]. Similar results have been observed in the case of patients with brain tumors [26]. This may indicate the strong utility of detecting the presence of cancer cells, regardless of their location.

Our working laboratory group’s study also showed that patients with EnOC (2733.13 pg/mL) had higher levels of MMP-11 compared to healthy women (233.13 pg/mL). This agrees with the study of Kicman et al. [1], who also found higher MMP-11 levels in ovarian cancer patients (1.50 ng/mL (1500 pg/mL), wherein EnOC accounted for 25% of all patients) compared to healthy women (0.24 ng/mL = 240 pg/mL). Other researchers have also confirmed our results, as they found that the concentration of MMP-11 was significantly elevated in OC patients [27]. However, in this work, the histopathological type of OC was not specified. At the same time, both in our study and in the study by Kicman et al. [1], patients with cysts had lower MMP-11 levels (190.0 pg/mL) than the other two study groups. Clinically, elevated MMP-11 levels in patients with EnOC may indicate increased activity of extracellular matrix remodeling processes and tumor progression, consistent with the role of this metalloproteinase in regulating the tumor microenvironment and facilitating tumor invasion. The observation of lower MMP-11 levels in patients with cysts suggests that this marker may have the potential to differentiate benign from malignant lesions, although due to its relatively low specificity, its diagnostic use seems most justified in combination with other markers. Other researchers who have described studies on MMP-11 expression in ovarian cancer tissues have reported that MMP-11 had no significant effect on patient prognosis [28] or overall survival [24]. Interestingly, patients with other types of cancer exhibit elevated levels of MMP-11 compared to healthy patients—such results have been reported in patients with colon cancer [29] and uterine corpus endometrial carcinoma [30].

In our study, we found the highest concentrations of MMP-10 in patients with benign lesions (208.80 pg/mL) compared to cancer patients (160.35 pg/mL) and healthy women (59.51 pg/mL). Importantly, higher levels of MMP-10 were found in patients with serous cysts compared to patients with endometrioid cysts. These results indicate that MMP-10 shows preliminary potential both in differentiating benign from malignant lesions as a negative marker and as a marker for differentiating histologically different subtypes of ovarian cysts, however, this requires further research. Clinically, elevated MMP-10 levels in patients with benign lesions, particularly serous cysts, may reflect activation of processes related to tissue remodeling and local inflammation rather than necessarily indicating the presence of neoplastic lesions. This suggests that MMP-10 may act as a marker of degenerative and inflammatory processes rather than a specific cancer biomarker. We are currently the first research laboratory group to demonstrate the diagnostic utility of MMP-10 in endometrioid ovarian cancer. However, in the work of Wei et al. [31], in which the expression of MMP-10 mRNA was studied in ovarian cancer tissues of other histological types (serous and mucinous), MMP-10 expression was significantly up-regulated in ovarian cancer tissue compared with healthy ovarian tissues. These results are consistent with ours; however, in the aforementioned work, the expression of mRNA in tissues with benign changes was not assessed. Therefore, further studies on determining the relationship in the expression and concentration of this metalloproteinase in different histological types and types of neoplastic changes are necessary. Similar results have been demonstrated by Mariya et al. [32]. On the other hand, in the work of Ke et al. [33], the authors presented completely contradictory results in which both mRNA expression and MMP-10 concentration in tissues were higher in neoplasms compared to benign lesions. Therefore, this requires confirmation; however, the authors’ studies were based on samples from four OC and four BL patients, and the histological type of the lesions was not clearly defined. Moreover, these differences may also be related to the tested material itself, because it cannot be clearly confirmed if MMP-10 accumulates in neoplastic lesions, mainly in tissues, and it is not secreted into the bloodstream. Nevertheless, in the course of different cancers such as breast cancer, higher levels of MMP-10 were found in the cancer group compared to women with benign lesions and healthy women [34]. Differences in concentrations observed between different types of cancer may be related to the histological type of the cancer, patients menopausal status, or the influence of other metalloproteinases and may be helpful in differential diagnosis.

In the case of MMP-26, similarly to MMP-10, higher levels of this enzyme were found in patients with ovarian cysts (10,985 pg/mL) compared to patients with endometrioid ovarian cancer (9425 pg/mL) and healthy women (7163 pg/mL). For MMP-26, no significant differences were found between endometrioid and serous cysts. This indicates that it does not show potential in the differential diagnosis of histological subtypes of ovarian cysts; however, it can be considered as a negative marker to differentiate benign from malignant lesions. Clinically elevated MMP-26 levels in patients with ovarian cysts may indicate the enzyme’s involvement in extracellular matrix remodeling and the local inflammatory response associated with benign lesions. The results obtained by our working laboratory group are in agreement with those of Kicman et al. [1], who also showed that MMP-26 levels were highest in patients with benign lesions (10.44 ng/mL) compared to patients with ovarian cancer (9.33 ng/mL) and healthy women (7.16 ng/mL). This indicates the reproducibility of the study. However, it should be noted that the research group of Kicman et al. [1] studied patients with different types of ovarian cancer. Also, the expression of MMP-26 was evaluated by Ripley et al. [35], who found increased expression of this metalloproteinase in ovarian cancer tissues compared to healthy tissues. MMP-26 immunostaining intensity increased with tumor stage, with invading tumor cells showing the strongest reaction. On the other hand, the study by Zhao et al. [36] revealed that MMP-26 tissue expression is not elevated in ovarian cancer tissues, which is in contrast with our results. However, this discrepancy might be connected with different study materials, as this MMP might be mainly secreted to the bloodstream, not cumulated in the cancer tissue itself. Also, increased expression of MMP-26 has been observed in endometrial cancer relative to healthy tissues, with it being higher with advanced tumor stage and depth of invasion [37].

Due to the fact that we did not obtain statistical significance for MMP-3, it was omitted in the Discussion section; however, there are works available that describe statistically significant differences for this parameter obtained for patients in the mixed OC group [1]. This may be related to the histological type, because in the case of our studies on EnOC, MMP-3 does not change, which requires further confirmation.

CA125 and HE4 concentrations were higher in the EnOC group compared to the EC and the HW groups. This is consistent with the results of other researchers and indicates the proper standardization of the applied methodology of our studies, compliance with GLP (Good Laboratory Practice), and proper selection of patients for the study group and control groups by maintaining appropriate exclusion criteria [1,2,18].

To determine potential correlations between the MMPs studied, we performed Spearman analysis. In the group of healthy women and EnOC patients, we showed no statistically significant correlations. In the group of women with benign lesions, we showed a number of statistically significant negative and positive correlations—between MMP-3, MMP-10, MMP-11, and routine markers. This indicates the dynamics of the interrelation of these markers in OC. We found only two studies that described correlations between the tested MMPs in a similar way. In the first one, the authors revealed a significant but weak positive correlation between MMP-11 and HE4 and the ROMA algorithm in women with OC [1]. In the second work, the authors established a significant weak correlation between MMP-7 and HE4, as well as MMP-7 and CA125 in OC patients and MMP-7 and HE4 in benign ovarian lesions [17]. However, they studied patients with other histological types and this may be related to the observed differences between the described work and our work.

Sensitivity, diagnostic specificity, diagnostic accuracy, and negative and positive predictive value were calculated to determine diagnostic utility. The MMPs tested exceeded the diagnostic parameter values obtained for routine markers. The highest values of diagnostic parameters were obtained for MMP-7 (SE: 98.55%; SP: 86% NPV: 87.72%; PPV: 95.41%; ACC: 57.58%); these results are comparable and even higher than the values of diagnostic parameters obtained by Będkowska et al. [17] in a mixed OC group (46% of EnOC), wherein authors established values for MMP-7 as follows: SE: 61%; SP: 95%; NPV: 93%; PPV: 61%. We also found one work that describes the diagnostic utility values for MMP-11 and MMP-26 in patients with ovarian cancer [1]. However, similarly to the work of Będkowska et al. [17], it was a mixed group of OC, where EnOC constituted 25% of patients. The authors obtained the following values of SE, 70.83%; SP: 76.00%; PPV: 87.63%; NPV: 52.05% for MMP-11, and SE: 78.33%; SP: 68.00%; PPV: 85.45%; NPV: 56.67% for MMP-26, which were comparable or lower than those obtained by us [17]. When comparing the diagnostic performance of the investigated MMPs with routinely applied markers such as CA125 and HE4, several important differences emerge. MMP-7 demonstrated the highest sensitivity (98.55%), even slightly surpassing CA125 (96%), while also showing superior specificity (86% vs. 47.95% for CA125 and 52.05% for HE4). This suggests that MMP-7 may provide a more balanced diagnostic profile than the established markers. Similarly, MMP-10 and MMP-26 reached sensitivities above 90%, comparable to CA125, but with improved specificity (62% and 60%, respectively), which could reduce the risk of false positives. In contrast, CA125, while achieving excellent PPV (97.96%) and accuracy (97%), remains limited by its low specificity, reflecting its known lack of disease-specificity in ovarian cancer diagnostics. HE4 showed moderate sensitivity (80%) with higher specificity than CA125 (52.05%), but its overall performance was still inferior to that of MMP-7. These findings indicate that especially MMP-7, and to a lesser extent MMP-10 and MMP-26, may complement or potentially improve upon the diagnostic utility of traditional markers, warranting further investigation in larger patient cohorts. To the best of our knowledge, these are the only currently available data addressing diagnostic performance parameters of MMPs in ovarian cancer. This emphasizes the novelty of our findings, as previous studies have not provided such a comprehensive diagnostic assessment. Consequently, our work represents one of the very few contributions in this field for ovarian cancer and the first to specifically investigate these parameters in endometrioid ovarian carcinoma. These results may therefore offer a foundation for further studies exploring the clinical utility of MMPs as complementary biomarkers alongside established markers such as CA125.

A final analysis to determine the power of the test was performed using ROC-AUC curves. In our study, the highest AUC values were obtained for MMP-7 (0.9658) from all tested parameters, which indicates the strong potential of MMP-7 in the diagnosis of EnOC, but it was slightly lower that established for CA125. Next were MMP-11 (0.7926), MMP-10 (0.7763), and MMP-26 (0.7708). Similar results were obtained by Będkowska et al. [17], where MMP-7 revealed AUC = 0.8335 (CA125: 0.8998; HE4: 0.8836) and Kicman et al. [1], where MMP-11 revealed AUC = 0.7022 and MMP-26 revealed AUC = 0.7751 (CA125: 0.9918; HE4: 0.9429). This may indicate that diagnostic usefulness may not change depending on the histological type of cancer. We did not find any studies describing the diagnostic usefulness of other MMPs, which makes our work extremely exceptional.

We would like to emphasize that this is the first study involving selected matrilysin and stromelysin family members as novel diagnostic biomarkers in a rare histological type of ovarian cancer, i.e., endometrioid ovarian cancer. It should be noted that MMP-7, MMP-10, MMP-11, and MMP-26 demonstrated favorable diagnostic utility values and show preliminary potential both as differentiation markers and as candidates for early diagnosis.

The clinical utility of individual MMPs, such as MMP-7 and MMP-10, should be considered in the context of established markers like CA125. Although these MMPs show high sensitivity and specificity, their overall accuracy does not surpass that of CA125. Given the small patient cohort, definitive conclusions cannot be drawn, and further studies in larger populations are needed. These markers may serve as complementary tools to enhance diagnostic precision rather than as standalone clinical markers.

An additional limitation lies in the fact that all patients were recruited from a single clinical center, which may limit the generalizability of the findings. Therefore, independent studies conducted in diverse populations and clinical settings are warranted to validate our preliminary results and to further assess the diagnostic performance of the most promising biomarkers identified in this study. Despite these constraints, our findings suggest that this research holds considerable promise for advancing the biochemical diagnosis of ovarian cancer.

What is worth mentioning is that matrix metalloproteinases can be present not only in the circulation in a soluble form but also on the surface of extracellular vesicles. In this vesicle-associated form, they are believed to be more resistant to self-degradation in the bloodstream, which may prolong their biological activity. Previous studies have also indicated that vesicle-bound metalloproteinases may reflect important aspects of tumor biology, and their presence has been linked to clinical and morphological parameters in serous ovarian cancer. For example, Yunusova et al. [38] demonstrated correlations between metalloproteinases expressed on the surface of small extracellular vesicles and both ascites volume and peritoneal carcinomatosis index in patients with advanced ovarian cancer. Nevertheless, it should be emphasized that the present study focuses on other metalloproteinases, and it would be of particular interest to also conduct similar investigations with enzymes beyond MMP-2 and MMP-9.

## 5. Conclusions

Although CA125 and HE4 remain the gold-standard serologic markers in ovarian cancer diagnostics, their limitations (particularly the low specificity of CA125) justify the search for novel biomarkers. In this context, MMP-7, MMP-10, MMP-11, and MMP-26 demonstrated promising diagnostic profiles in our cohort, suggesting that, rather than replacing established markers, they may serve as valuable complementary tools to improve diagnostic precision, especially in endometrioid ovarian cancer, and as markers in the differentiation of benign from malignant lesions.

## Figures and Tables

**Figure 1 biomedicines-13-02143-f001:**
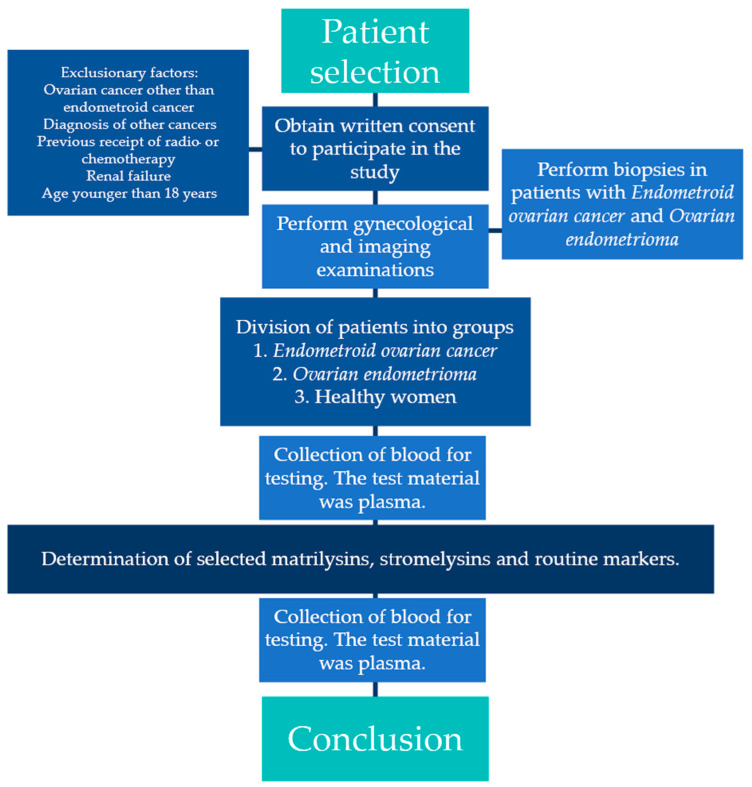
Flowchart of the experiment.

**Figure 2 biomedicines-13-02143-f002:**
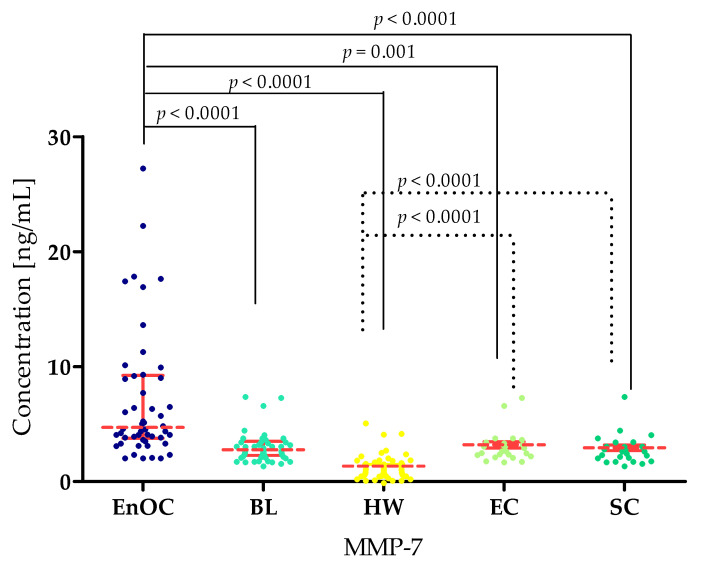
MMP-7 concentrations in the study groups—*Endometrioid Ovarian carcinoma* (EnOC), benign lesions (BL), healthy women (HW), endometrial cysts (EC), and serous cysts (SC).

**Figure 3 biomedicines-13-02143-f003:**
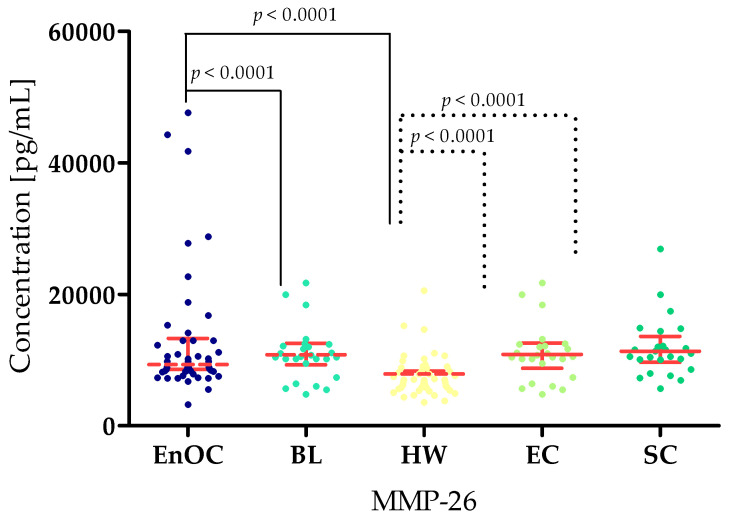
MMP-26 concentrations in the study groups—*Endometrioid Ovarian carcinoma* (EnOC), benign lesions (BL), healthy women (HW), endometrial cysts (EC) and serous cysts (SC).

**Figure 4 biomedicines-13-02143-f004:**
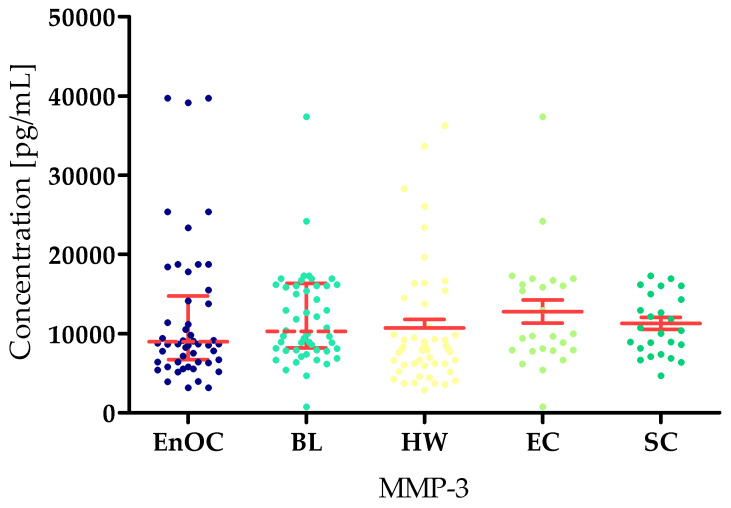
MMP-3 concentrations in the study groups—*Endometrioid Ovarian carcinoma* (EnOC), benign lesions (BL), healthy women (HW), endometrial cysts (EC), and serous cysts (SC).

**Figure 5 biomedicines-13-02143-f005:**
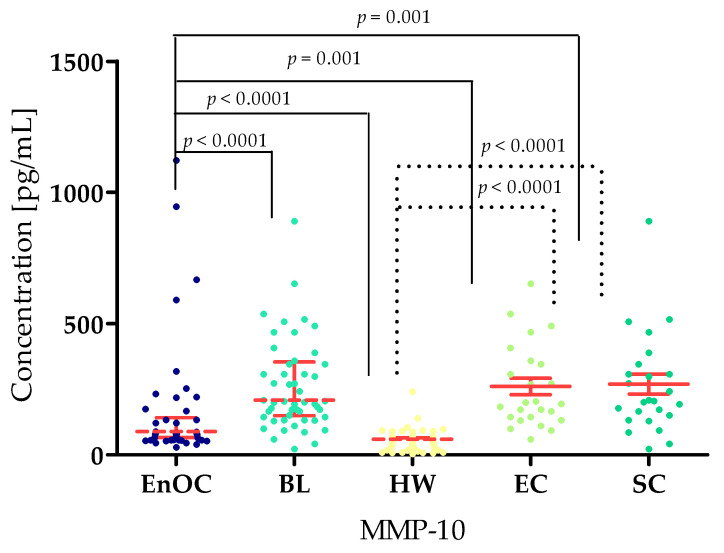
MMP-10 concentrations in the study groups—*Endometrioid Ovarian carcinoma* (EnOC), benign lesions (BL), healthy women (HW), endometrial cysts (EC), and serous cysts (SC).

**Figure 6 biomedicines-13-02143-f006:**
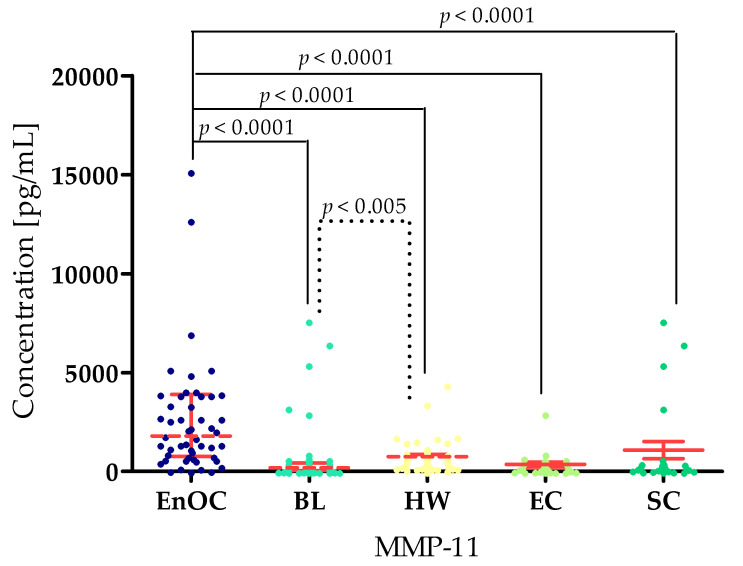
MMP-11 concentrations in the study groups—*Endometrioid Ovarian carcinoma* (EnOC), benign lesions (BL), healthy women (HW), endometrial cysts (EC), and serous cysts (SC).

**Figure 7 biomedicines-13-02143-f007:**
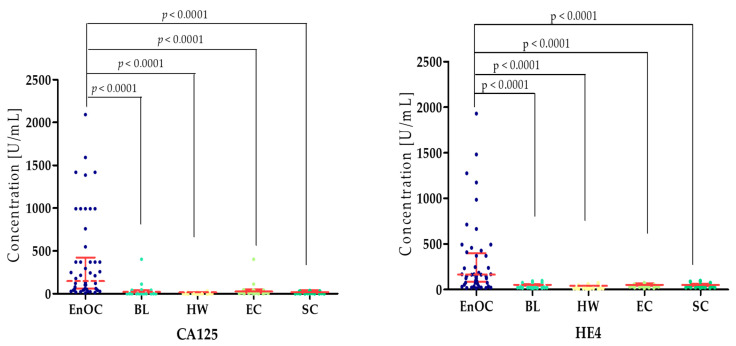
CA125 and HE4 concentrations in the study groups—*Endometrioid Ovarian carcinoma* (EnOC), benign lesions (BL), healthy women (HW), endometrial cysts (EC), and serous cysts (SC).

**Figure 8 biomedicines-13-02143-f008:**
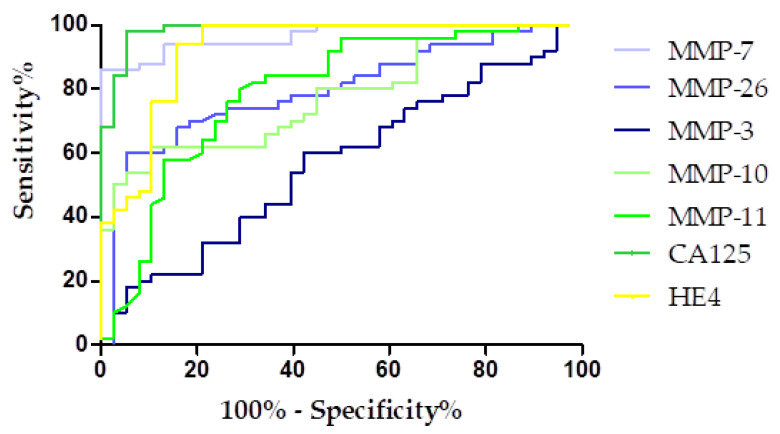
ROC curve for tested parameters.

**Table 1 biomedicines-13-02143-t001:** Characteristics of examined groups: ovarian *endometrioid carcinoma,* benign lesions *(endometrial cysts* and *serous cysts),* and healthy subjects.

Ovarian *Endometrioid Carcinoma (EnOC)*	
Number of patients—50 (100%) Median of age: 56 Menopausal status Pre-menopause—12 Postmenopause—38 Tumor stage I–II—25 III–IV—25	
Ovarian Benign Lesions (BL)	
*Endometrial Cysts (EC)*	*Serous Cysts (SC)*
Number of patients—25 (50%) Median of age: 44 Menopausal status Pre-menopause—16 Postmenopause—9	Number of patients—25 (50%) Median of age: 56 Menopausal status Pre-menopause—5 Postmenopause—20
Healthy Women (HW)	
Number of patients—50 (100%) Median of age: 60 Menopausal status Pre-menopause—24 Postmenopause—26	

**Table 2 biomedicines-13-02143-t002:** Plasma concentrations of the studied parameters in the tested groups (EnOC and HW).

Endometrioid *Ovarian Carcinoma (EnOC)*							
Parameter	MMP-7 [ng/mL]	MMP-26 [pg/mL]	MMP-3 [pg/mL]	MMP-10 [pg/mL]	MMP-11 [pg/mL]	CA125 [U/mL]	HE4 [U/mL]
Median	5.64	9425.00	12,106.47	160.35	2733.13	149.10	338.01
Standard deviation	5.93	7332.16	8519.96	190.39	3138.24	496.78	410.71
Minimum	2.20	3578.00	3468.52	38.90	40.00	19.60	35.57
Quartile 1	3.77	8625.00	6705.05	65.32	766.75	64.87	90.52
Quartile 3	9.64	13,310.00	14,768.24	177.36	3900.00	609.55	430.30
Maximum	27.40	42,070.00	40,000.00	954.50	15,180.00	2100.00	1944.00
Healthy Women (HW)							
Parameter	MMP-7 [ng/mL]	MMP-26 [pg/mL]	MMP-3 [pg/mL]	MMP-10 [pg/mL]	MMP-11 [pg/mL]	CA125 [U/mL]	HE4 [U/mL]
Median	0.95	7163.00	10,719.41	59.51	762.00	16.90	39.51
Standard deviation	1.08	7903.60	7605.36	41.96	807.22	7.45	14.68
Minimum	0.04	3037.55	3178.22	12.30	20.00	4.24	23.52
Quartile 1	0.66	3920.00	6329.89	28.37	260.00	11.38	31.07
Quartile 3	1.87	7162.50	11,880.69	77.06	895.00	22.27	44.53
Maximum	5.23	9128.75	36,528.70	249.54	4400.00	39.94	1944.20

**Table 3 biomedicines-13-02143-t003:** Plasma concentrations of the studied parameters in the tested groups (endometrial cysts and serous cysts).

Benign Lesions Total Group (BL)							
Parameter	MMP-7 [ng/mL]	MMP-26 [pg/mL]	MMP-3 [pg/mL]	MMP-10 [pg/mL]	MMP-11 [pg/mL]	CA125 [U/mL]	HE4 [U/mL]
Median	2.78	10,985.00	10,298.00	208.80	190.00	23.38	52.55
Standard deviation	1.28	4342.00	10,298.00	170.50	1610.00	57.43	18.61
Minimum	1.50	5140.00	1054.00	30.98	20.00	8.900	28.20
Quartile 1	2.27	9591.00	8211.00	149.70	55.00	13.13	43.65
Quartile 3	3.50	12,539.00	16,332.00	353.90	440.00	43.98	62.91
Maximum	7.52	27,235.00	37,648.00	898.10	7640.00	410.30	112.30
Ovarian *Endometrioid Cyst* Group *(EC)*							
Parameter	MMP-7 [ng/mL]	MMP-26 [pg/mL]	MMP-3 [pg/mL]	MMP-10 [pg/mL]	MMP-11 [pg/mL]	CA125 [U/mL]	HE4 [U/mL]
Median	2.80	10,825.00	10,235.00	160.35	233.13	24.98	53.20
Standard deviation	3.17	11,265.58	12,604.60	190.39	1338.24	496.78	410.71
Minimum	1.31	4122.06	7245.47	38.90	40.00	19.60	35.57
Quartile 1	1.84	5140.00	1054.22	65.32	76.75	64.87	90.52
Quartile 3	2.79	10,825.00	9968.57	177.36	390.00	609.55	430.30
Maximum	3.51	12,538.75	16,615.94	954.48	7518.00	2100.00	1944.20
Ovarian *Serous Cyst* Group *(SC)*							
Parameter	MMP-7 [ng/mL]	MMP-26 [pg/mL]	MMP-3 [pg/mL]	MMP-10 [pg/mL]	MMP-11 [pg/mL]	CA125 [U/mL]	HE4 [U/mL]
Median	2.94	12,169.00	10,683.00	113.80	200.00	19.00	51.47
Standard deviation	1.25	4519.00	3774.00	187.10	1160.00	13.54	22.46
Minimum	1.50	5975.00	4972.00	30.98	20.00	9.00	35.15
Quartile 1	2.07	9660.00	8079.00	150.50	90.00	12.58	40.86
Quartile 3	3.39	13,593.00	14,959.00	335.30	430.00	37.10	62.95
Maximum	7.52	27,235.00	17,599.00	898.10	7640.00	47.30	112.30

**Table 4 biomedicines-13-02143-t004:** Spearman correlation in the BL total group and SC group (only significant results).

	MMP-7	MMP-26	MMP-3	MMP-10	MMP-11	CA125	HE4
	BL Total Group						
MMP-3	N/S	N/S	-	N/S	*p* = 0.0238 r = 0.4420	*p* = 0.0210 r = −0.1591	*p* = 0.0100 r = −0.2545
MMP-10	N/S	N/S	N/S	-	*p* = 0.0490 r = −0.2904	N/S	N/S
MMP-11	*p* = 0.0410 r = 0.4506	N/S	*p* = 0.0238 r = 0.4420	*p* = 0.0160 r = −0.4675	-	*p* = 0.0145 r = −0.4358	*p* = 0.0105 r = −0.4660
	SC Group						
MMP-26	N/S	-	N/S	*p* < 0.0001 r = −0.7065	*p* < 0.0001 r = −0.6291	*p* < 0.0001 r = −0.6602	*p* < 0.0001 r = −0.7110
MMP-3	*p* < 0.0001 r = −0.7041	*p* = 0.0088 r = 0.3668	-	*p* < 0.0001 r = −0.6660	*p* < 0.0001 r = −0.6664	*p* < 0.0001 r = −0.6564	*p* < 0.0001 r = 0.6743
MMP-10	N/S	N/S	*p* < 0.0001 r = −0.6256	-	*p* < 0.0001 r = −0.6423	*p* < 0.0001 r = −0.7456	*p* < 0.0001 r = −0.7114
MMP-11	*p* = 0.0089 r = 0.3552	N/S	*p* = 0.0084 r = 0.3447	*p* = 0.0081 r = 0.3657	N/S	*p* < 0.0001 r = −0.8542	*p* < 0.0001 r = −0.5475

Abbreviations: N/S—non significant.

**Table 5 biomedicines-13-02143-t005:** Diagnostic criteria for MMPs, CA125, and HE4.

Parameter	SE [%]	SP [%]	PPV [%]	NPV [%]	ACC [%]
MMP-7	98.55%	86.00%	87.72%	95.41%	57.58%
MMP-26	94.00%	60.00%	70.15%	90.91%	77.00%
MMP-3	58.00%	60.00%	59.18%	58.82%	59.00%
MMP-10	92.00%	62.00%	70.77%	88.57%	77.00%
MMP-11	32.00%	82.00%	64.00%	54.67%	57.00%
CA125	96.00%	47.95%	97.96%	96.08%	97.00%
HE4	80.00%	52.05%	93.02%	82.46%	87.00%

**Table 6 biomedicines-13-02143-t006:** Diagnostic power of tests by ROC function of the tested parameters and tumor markers.

Parameters	MMP-7	MMP-26	MMP-3	MMP-10	MMP-11	CA125	HE4
AUC	0.9658	0.7708	0.5679	0.7763	0.7926	0.9758	0.9247
SE_AUC_	0.0165	0.0520	0.0616	0.0485	0.0509	0.0101	0.0306
95% CI	0.9335–0.9981	0.6688–0.8727	0.4471–0.6886	0.6812–0.8714	0.6928–0.8925	0.9660–1.006	0.8647–0.9848
*p* (AUC = 0.5)	<0.0001	<0.0001	0.2772	<0.0001	<0.0001	<0.0001	<0.0001

## Data Availability

The raw data supporting the conclusions of this article will be made available by the authors upon request.

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
