# Peer review of "Diagnostic Use of Selected Metalloproteinases in Endometrioid Ovarian Cancer"

_biomedicines, 2025, doi:10.3390/biomedicines13092143_

Round 1

Reviewer 1 Report

Comments and Suggestions for Authors

The study is characterized by a careful selection of clinical material. The level of seven tumor markers was analyzed, including five markers belonging to the family of secreted matrix metalloproteinases (MMPs).

There are a number of comments or questions that require clarification regarding the work submitted for review  of the original paper “Diagnostic Use of Selected Metalloproteinases in Endometrioid    2

Ovarian Cancer “ by  Gacuta Ewa  et al.

  1. Page 3 of the manuscript. It is written that some patients of the main group underwent computed tomography and magnetic resonance imaging of the pelvic organs. In accordance with international clinical guidelines, absolutely all patients with suspected ovarian cancer should undergo these studies for diagnostic purposes, as well as to clarify the prevalence of the tumor process and plan treatment - it is necessary to correct.
  2. Chapter Results - there is no need to duplicate the results in tables and figures - it is enough to leave the figures
  3. Tables 4 and 5 are not informative. It is enough to present in them only statistically significant Spearman correlations at p < 0.05 - these tables need to be redone.
  4. In my opinion, in this work, in the Discussion chapter, it is logical to note that secreted matrix metalloproteinases can be found not only in the blood in soluble form, but also as part of extracellular vesicles on their crown. A number of authors believe that in this form they are more resistant to their own degradation in the bloodstream. And some of them have already been used in association with clinical and morphological parameters to predict serous ovarian cancer (for example, Yunusova NV, Patysheva MR, Molchanov SV, Zambalova EA, Grigor'eva AE, Kolomiets LA, Ochirov MO, Tamkovich SN, Kondakova IV. Metalloproteinases at the surface of small extrcellular vesicles in advanced ovarian cancer: Relationships with ascites volume and peritoneal canceromatosis index. Clin Chim Acta. 2019 Jul;494:116-122. doi: 10.1016/j.cca.2019.03.1621).

Conclusion: Revision and re-review

Author Response

The study is characterized by a careful selection of clinical material. The level of seven tumor markers was analyzed, including five markers belonging to the family of secreted matrix metalloproteinases (MMPs). There are a number of comments or questions that require clarification regarding the work submitted for review of the original paper “Diagnostic Use of Selected Metalloproteinases in Endometrioid Ovarian Cancer“ by Gacuta Ewa et al.

Dear Reviewer, we would like to thank You very much for your thorough and honest review of our manuscript “Diagnostic Use of Selected Metalloproteinases in Endometrioid Ovarian Cancer”. We will try to carefully answer all the Reviewer's questions and objections.

Comment 1: Page 3 of the manuscript. It is written that some patients of the main group underwent computed tomography and magnetic resonance imaging of the pelvic organs. In accordance with international clinical guidelines, absolutely all patients with suspected ovarian cancer should undergo these studies for diagnostic purposes, as well as to clarify the prevalence of the tumor process and plan treatment - it is necessary to correct.

Response 1: We apologize for this mistake and thank the reviewer for pointing it out. The text has been corrected to reflect that, in accordance with international clinical guidelines, all patients with suspected ovarian cancer should undergo computed tomography and magnetic resonance imaging (lines 105-106).

Comment 2: Chapter Results - there is no need to duplicate the results in tables and figures - it is enough to leave the figures

Response 2: Thank you for the comment. However, we would prefer to keep both the figures and the tables in the manuscript. Our reasoning is that the figures do not fully convey the numerical values presented in the tables, while the tables provide more detailed data than the figures. We kindly ask for the reviewer’s understanding in this regard.

Comment 3: Tables 4 and 5 are not informative. It is enough to present in them only statistically significant Spearman correlations at p < 0.05 - these tables need to be redone.

Response 3: Thank you for the comment. Table 4 has been removed due to the lack of statistically significant results, while Table 5 has been appropriately revised. Only statistically significant values are now presented.

Comment 4: In my opinion, in this work, in the Discussion chapter, it is logical to note that secreted matrix metalloproteinases can be found not only in the blood in soluble form, but also as part of extracellular vesicles on their crown. A number of authors believe that in this form they are more resistant to their own degradation in the bloodstream. And some of them have already been used in association with clinical and morphological parameters to predict serous ovarian cancer (for example, Yunusova NV, Patysheva MR, Molchanov SV, Zambalova EA, Grigor'eva AE, Kolomiets LA, Ochirov MO, Tamkovich SN, Kondakova IV. Metalloproteinases at the surface of small extrcellular vesicles in advanced ovarian cancer: Relationships with ascites volume and peritoneal canceromatosis index. Clin Chim Acta. 2019 Jul;494:116-122. doi: 10.1016/j.cca.2019.03.1621).

Response 4: Thank you for this valuable comment. We agree that it is important to emphasize in the Discussion that matrix metalloproteinases can be detected not only in soluble form in the bloodstream, but also on the surface of extracellular vesicles. This vesicle-associated form is considered to be more resistant to self-degradation in circulation, which may prolong their activity. We have now added this aspect to the Discussion (lines 512-523).

Again, we thank the reviewer for all the guidance and corrections. It is our hope that the manuscript, after revision, will meet the reviewer's expectations and be published in the Biomedicines.

Reviewer 2 Report

Comments and Suggestions for Authors

The manuscript entitled "Diagnostic Use of Selected Metalloproteinases in Endometrioid Ovarian Cancer" was revised very carefully. The manuscript has merit, but the authors' presentation of the discussion does not directly relate to the main objective: "the present study aimed to evaluate the concentrations and diagnostic utility of selected matrilysin and stromelysin en-80 enzymes in EnOC.". In fact, the objective of their study is more fitting for its title.

Authors must attend minor and major observations:

Major observations:

Authors must have to discuss all results of Table 6 contain the diagnostic criteria – diagnostic sensitivity (SE), diagnostic spec-240 ificity (SP), positive and negative predictive value (PPV and NPV), and accuracy (ACC) in 241 patients with ovarian endometrioid carcinoma: it is the central part of the objective of the manuscript. And contrast this with literature data, at least regarding CA125.

For example, what do you mean MMP-7 has higher SP, SE, but lower ACC and PPV, versus CA125 and HE4, which could represent the "golden standard serologic markers"? It could be selected as in conclusions, authors say: "MMP-7, MMP-10, MMP-11 and MMP-26 show potential as biomarkers in the diagnostics of endometrioid ovarian cancer, and as markers to differentiate benign from malignant lesions"

Also,

What do you mean MMP-10 has low SP, SE, ACC, and PPV, versus CA125, which is higher in all of them, which could represent the "golden standard serologic markers"? It could be selected as in conclusions, authors say: "MMP-7, MMP-10, MMP-11 and MMP-26 show potential as biomarkers in the diagnostics of endometrioid ovarian cancer, and as markers to differentiate benign from malignant lesions"

What do you mean by MMP-11 having higher SP but lower values in SE, ACC, and PPV, compared to CA125, which is higher in all of these? Could this represent the "golden standard serologic markers"? MMP-11 could be selected as in conclusions, authors say: "MMP-7, MMP-10, MMP-11 and MMP-26 show potential as biomarkers in the diagnostics of endometrioid ovarian cancer, and as markers to differentiate benign from malignant lesions"

Authors must discuss the clinical implications of considering one MMP, if someone could be selected as a clinical marker or not, versus levels of the gold standard, CA-125, or another marker.

Authors must discuss all these parameters in terms of clinical significance, rather than merely discussing the results in the literature if some of these MMPs are lower or higher than their results. And authors have to revise in the discussion about this argument, when a data from literature is "gene regulation", from "protein expression", because sometimes, they never take care of this important point.

Minor observations:

  1. In abstract, authors never say where these MMPs were quantified, serum, Plasma?. At the start of the diagnosis?
  2. "Currently, the number of compounds with high potential in the diagnosis of ovarian cancer is steadily increasing."; mention examples and indicate references.
  3. "However, enzymes from the 69 matrix metalloproteinases (MMPs) group are of particular interest." Explain the reason why MMPs are of particular interest.
  4. In Table 1, revise the total number is 55 patients from a sum of pre- and post-menopausic; and the number must be 50
  5. "(1.50ng/mL (1500 pg/ml), it does not make sense, maybe authors are referring to 1500 pg/microliter?
  6. "We are currently the first research team to demonstrate the diagnostic utility of MMP-10"; what is the evidence to say this?. If the AUC was higher for MMP-7 and MMP-11
  7. "n work of Wei et al. [27] in which the expression of MMP-11 mRNA was studied in ovarian cancer tissues of other histological types (serous and mucinous), MMP-10 expression were significantly up-regulated in ovarian cancer tissue compared with normal ovarian tissues. These results are consistent with ours, however, in the aforementioned work the expression of mRNA"; authors must cite articles with information about protein levels, not mRNA. Please revise.
  8. "Results obtained by our team", our working Laboratory group. It is not a game-play
  9. There are several fragments without references, as:
  • "EnOC 62 is a distinct histological subtype with unique molecular and clinical characteristics, distin-63 guishing it from other epithelial ovarian cancers such as serous or clear cell carcinoma."
  • "Their increased activity is mostly associated with the initiation and progression of many diseases, but a special role is attributed to them in the process of carcinogenesis. At the initiation stage, they are associated with the induction of genomic instability whereas at subsequent stages, their activity is associated with an increase in the frequency of proliferation, invasion and migration of tumor cells and the induction of angiogenesis"

Author Response

The manuscript entitled "Diagnostic Use of Selected Metalloproteinases in Endometrioid Ovarian Cancer" was revised very carefully. The manuscript has merit, but the authors' presentation of the discussion does not directly relate to the main objective: "the present study aimed to evaluate the concentrations and diagnostic utility of selected matrilysin and stromelysin en-80 enzymes in EnOC.". In fact, the objective of their study is more fitting for its title.

Dear Reviewer, thank you very much for your important comments regarding our manuscript. We would like to thank you for such an accurate analysis. All your recommendations were extremely helpful and offered us the opportunity to significantly improve this paper.

Comment 1:  Authors must have to discuss all results of Table 6 contain the diagnostic criteria – diagnostic sensitivity (SE), diagnostic specificity (SP), positive and negative predictive value (PPV and NPV), and accuracy (ACC) in patients with ovarian endometrioid carcinoma: it is the central part of the objective of the manuscript. And contrast this with literature data, at least regarding CA125.

Response 1: We thank the Reviewer for this valuable comment. The results presented in Table 6 regarding diagnostic sensitivity (SE), specificity (SP), positive predictive value (PPV), negative predictive value (NPV), and accuracy (ACC) in patients with ovarian endometrioid carcinoma have been discussed in detail in corrected version of the manuscript. In this section, we contrasted our findings with the routinely used laboratory diagnostic markers, including CA125, which served as a comparative reference. Importantly, to the best of our knowledge, the studies we have cited and compared are the only ones available in the literature that address this topic. We were not able to identify additional publications reporting diagnostic performance parameters of MMPs in ovarian cancer, which makes our study one of the very few available for OC, and the first to address EnOC specifically.

Comment 2:  For example, what do you mean MMP-7 has higher SP, SE, but lower ACC and PPV, versus CA125 and HE4, which could represent the "golden standard serologic markers"? It could be selected as in conclusions, authors say: "MMP-7, MMP-10, MMP-11 and MMP-26 show potential as biomarkers in the diagnostics of endometrioid ovarian cancer, and as markers to differentiate benign from malignant lesions"

Response 2: We appreciate the reviewer’s insightful comment. Indeed, CA125 and HE4 remain the current gold-standard serologic markers in ovarian cancer diagnostics, as reflected by their high overall diagnostic accuracy and predictive values in our analysis. However, both markers have important limitations, most notably the relatively low specificity of CA125, which often leads to false-positive results, particularly in benign gynecological conditions. In our study, MMP-7 demonstrated remarkably high sensitivity (98.55%) and specificity (86.00%), exceeding those of CA125 and HE4, although its overall accuracy and PPV were lower. This apparent discrepancy may be partly explained by the limited sample size and the distribution of benign versus malignant cases in our cohort. It also suggests that while MMP-7 alone may not outperform established markers in terms of accuracy, its performance profile (especially the combination of very high SE and SP) indicates that it could serve as a valuable complementary marker. Therefore, our conclusions do not suggest that MMPs should replace CA125 or HE4 as diagnostic standards. Rather, we propose that markers such as MMP-7, MMP-10, MMP-11, and MMP-26 hold potential as adjunctive tools that, when combined with existing markers, may improve the differentiation between benign and malignant lesions and enhance diagnostic precision in endometrioid ovarian cancer, which was corrected in new version of our manuscript.

Comment 3:  What do you mean MMP-10 has low SP, SE, ACC, and PPV, versus CA125, which is higher in all of them, which could represent the "golden standard serologic markers"? It could be selected as in conclusions, authors say: "MMP-7, MMP-10, MMP-11 and MMP-26 show potential as biomarkers in the diagnostics of endometrioid ovarian cancer, and as markers to differentiate benign from malignant lesions"

Response 3: We agree with the Reviewer that (similarly to the case of MMP-7) CA125 remains superior to MMP-10 in terms of specificity, accuracy, and PPV. Our aim was not to claim that MMP-10 alone is a better diagnostic tool than CA125, but rather that it may provide supportive information when combined with established markers. To avoid any misinterpretation, the conclusions have been revised accordingly, in line with the modifications described in the previous point.

Comment 4:  What do you mean by MMP-11 having higher SP but lower values in SE, ACC, and PPV, compared to CA125, which is higher in all of these? Could this represent the "golden standard serologic markers"? MMP-11 could be selected as in conclusions, authors say: "MMP-7, MMP-10, MMP-11 and MMP-26 show potential as biomarkers in the diagnostics of endometrioid ovarian cancer, and as markers to differentiate benign from malignant lesions"

Response 4: We agree with the Reviewer. Similarly to MMP-7 and MMP-10, CA125 remains superior to MMP-11 in terms of SE, ACC, and PPV, while MMP-11 shows higher specificity. Our intention is not to suggest that MMP-11 surpasses established markers, but that it may provide complementary diagnostic information. This has been clarified in the revised conclusions.

Comment 5:  Authors must discuss the clinical implications of considering one MMP, if someone could be selected as a clinical marker or not, versus levels of the gold standard, CA-125, or another marker.

Response 5: We thank the Reviewer for this comment. We acknowledge that the clinical utility of individual MMPs must be interpreted in the context of established markers such as CA125. While some MMPs, particularly MMP-7 and MMP-10, demonstrate high sensitivity and specificity, their overall diagnostic accuracy and predictive values do not surpass those of CA125. Moreover, given the relatively small number of patients included in our study, it is difficult to draw definitive conclusions regarding the clinical applicability of a single MMP. Therefore, we propose that these MMPs should be considered complementary markers, potentially enhancing diagnostic precision when used alongside CA125 or other routine markers, rather than as standalone clinical markers. Further studies in larger cohorts are needed to confirm these findings.

Comment 6:  Authors must discuss all these parameters in terms of clinical significance, rather than merely discussing the results in the literature if some of these MMPs are lower or higher than their results. And authors have to revise in the discussion about this argument, when a data from literature is "gene regulation", from "protein expression", because sometimes, they never take care of this important point.

Response 6: We thank the Reviewer for this comment. The discussion has been revised to interpret all MMP parameters in terms of clinical significance and to clearly distinguish between data from gene regulation (mRNA) and protein expression, ensuring that clinical implications are addressed based on protein-level data where it was possible.

Comment 7:  In abstract, authors never say where these MMPs were quantified, serum, Plasma?. At the start of the diagnosis?

Response 7: Thank you, this has been corrected.

Comment 8:  "Currently, the number of compounds with high potential in the diagnosis of ovarian cancer is steadily increasing."; mention examples and indicate references.

Response 8: Thank you for this comment. We have revised the sentence to include specific examples and references. Currently, the number of compounds with high potential in the diagnosis of ovarian cancer is steadily increasing, including CA-125, HE4, mesothelin, and several matrix metalloproteinases (e.g., MMP-7 and MMP-11). These markers have been investigated for their diagnostic utility, alone or in combination, to improve early detection and patient stratification.

Comment 9:  "However, enzymes from the 69 matrix metalloproteinases (MMPs) group are of particular interest." Explain the reason why MMPs are of particular interest.

Response 9: Thank you for this comment. We have revised the text to clarify why MMPs are of particular interest. Matrix metalloproteinases (MMPs) are enzymes that degrade extracellular matrix components, playing a crucial role in tissue remodeling, inflammation, and tumor progression. They are of particular interest because their dysregulation has been linked to cancer development, metastasis, and other pathological processes, making them potential diagnostic biomarkers and therapeutic targets in oncology and metabolic diseases.

Comment 10:  In Table 1, revise the total number is 55 patients from a sum of pre- and post-menopausic; and the number must be 50

Response 10: Thank you, this has been corrected.

Comment 11:  "(1.50ng/mL (1500 pg/ml), it does not make sense, maybe authors are referring to 1500 pg/microliter?

Response 11: Thank you for this comment. We would like to clarify that 1.50 ng/mL is equivalent to 1500 pg/mL, and the values are consistent. The unit is correctly expressed as pg/mL, not pg/µL. The conversion in parentheses was included to unify the units used in two different studies cited in the manuscript, ensuring easier comparison for the reader.

Comment 12:  "We are currently the first research team to demonstrate the diagnostic utility of MMP-10"; what is the evidence to say this?. If the AUC was higher for MMP-7 and MMP-11

Response 12: Thank you for this comment. To the best of our knowledge, our study is the first to investigate the diagnostic utility of MMP-10 in endometrioid ovarian cancer (EnOC) and ovarian cancer (OC). We did not identify any previous publications reporting its diagnostic performance in these contexts in the available literature or databases. While the AUC values for MMP-10 are indeed lower than those for MMP-7 and MMP-11, our intention was not to claim that MMP-10 is the most effective marker, but rather to highlight that its diagnostic potential has not been previously described.

Comment 13:  "In work of Wei et al. [27] in which the expression of MMP-11 mRNA was studied in ovarian cancer tissues of other histological types (serous and mucinous), MMP-10 expression were significantly up-regulated in ovarian cancer tissue compared with normal ovarian tissues. These results are consistent with ours, however, in the aforementioned work the expression of mRNA"; authors must cite articles with information about protein levels, not mRNA. Please revise.

Response 13: Thank you for this comment. We would like to clarify that, to the best of our knowledge, there are currently no published studies investigating MMP-10 protein levels in ovarian cancer. The available literature reports only mRNA expression data, which is why we cited these studies. We acknowledge this limitation and have clarified it in the manuscript.

Comment 14:  "Results obtained by our team", our working Laboratory group. It is not a game-play

Response 14: Thank you, this has been corrected.

Comment 15:  There are several fragments without references, as:

"EnOC 62 is a distinct histological subtype with unique molecular and clinical characteristics, distin-63 guishing it from other epithelial ovarian cancers such as serous or clear cell carcinoma."

"Their increased activity is mostly associated with the initiation and progression of many diseases, but a special role is attributed to them in the process of carcinogenesis. At the initiation stage, they are associated with the induction of genomic instability whereas at subsequent stages, their activity is associated with an increase in the frequency of proliferation, invasion and migration of tumor cells and the induction of angiogenesis"

Response 15: Thank you. The references have been added.

In summary, we would like to thank you once again for such an accurate analysis of our manuscript and all the valuable recommendations. We hope that the paper in its revised and improved form is suitable for publication.

Round 2

Reviewer 2 Report

Comments and Suggestions for Authors

For the second time, the manuscript entitled "Diagnostic Use of Selected Metalloproteinases in Endometrioid Ovarian Cancer" was carefully revised. 

After doing that, this Reviewer wants to congratulate and thank the authors for attending to all the recommendations suggested and for doing so, taking care to present the utility of the MMPs analyzed. The text of the manuscript has been improved significantly. Thank you to the authors.

This Reviewer does not have any inconvenent in to proceed to accept the manuscript and publish it